# OpenReview forum: "Optimizing Visual Generative Models with Distribution-wise Rewards"
_ICLR.cc/2026/Conference — ICLR 2026 Conference Withdrawn Submission_

### Official Review · Reviewer_XS6c · 2025-10-20

**Soundness:** 3
**Presentation:** 2
**Contribution:** 2
**Rating:** 4
**Confidence:** 4

**Summary:**

Traditional RL methods for T2I generation offen suffer from the problem of reward-hacking, which optimizes each sample separately and disrupts the distribution of generated images. This paper introduces a subset-replace strategy that enables optimization with distribution-level rewards for generative models. In addition, the authors propose to optimize the model merging coefficients using the RL method, thereby enhancing post-hoc model optimization.

**Strengths:**

Intuitively, optimizing with distribution-level rewards is a sensible way to mitigate reward hacking. In this work, the authors propose the subset-replace strategy to reduce the computational cost of reward evaluation, enabling FID to serve as the RL reward signal. Building on the proposed RL method, the authors introduce RL-EMA to learn better model-merging coefficients in the post hoc procedure; RL-EMA can be applied to various methods.

**Weaknesses:**

- **Existing risks of information leakage of test data**: When measuring FID during the training process, one computes the distance between the reference image set and a target image set. When the target image set is constructed from the test dataset, information leakage can occur, making the evaluation unfair in comparison.

- **Enhancing the distribution of generated images towards a certain set using RL has limited practical value**: When users using a generative model, they may care about the text alignment, the aesthetic of generated samples, and so on. In my opinion,  that's why those sample-level rewards are proposed, which can score each sample from certain perspective, i.e., text following. Users won’t care about whether the generated image adhere to a certain distribution. The effectiveness of the proposed distribution-wise reward method on those more user-friendly metrics has not been explored.

- **Lacking statement of training cost**: It can be time-consuming when generating thousands of samples per step and calculating the FID metric at each step, and refreshing the reference set (generating 50K samples) with the current model every 10 steps. Even if the total number of training steps is small, the total training time can be huge. It will be more clear when the detailed training time is given.

**Questions:**

- In Figure 1, is the performance of the sample-wise RL method really that poor? There are now many successful sample-wise RL approaches; is there an issue with the choice of baseline?

- FID typically becomes more accurate as the number of evaluation samples increases. In Figure 3, theoretically, increasing the reference set size and the number of replaced images should yield better results, but this is not observed in practice. What is the reason for this?

---

### Official Review · Reviewer_U51e · 2025-10-27

**Soundness:** 1
**Presentation:** 2
**Contribution:** 3
**Rating:** 2
**Confidence:** 3

**Summary:**

- The authors propose two methods for post-training of image generative models that rely on evaluations of distributions of samples rather than just evaluations of individual samples. In particular, they leverage the FID as a distribution-wise reward and use a subset-replace strategy to get computationally tractable reward signal.
- For the first method, they replace a small subsets of images in some reference set with newly-generated images and compute the FID of this modified set. This value is then normalized into an advantage signal to train the network using either replacement sampling or policy gradient. Empirically, they show this method yields noticeable FID improvements to a pretrained SiT model.
- The second method consists of learning optimal weights for model merging. To do so, a small MLP is trained to generated model coefficients with a similar subset-replace strategy being used to provide reward signal to the MLP. The approach yields marginal improvements to EDM2-XS.
- Finally, the authors perform ablations, comparing the effect of the reference set size, the size of the replaced subset and sample selection strategies. Importantly, they notice a significant drop in performance when a model post-trained with SDE rollouts is evaluated using ODE inference.

**Strengths:**

- Non-trivial experiments testing the authors methods on large, near SOTA models.
- The first method is well-motivated and offers noticeable improvements to FID.
	- Qualitatively, the method does seem to fix artifacts/improve low quality images in multiple cases.
- The ablations help understand the impact of various design decisions.
- Training diffusion models with distribution-wise rewards makes sense and appears to be a novel idea.

**Weaknesses:**

- There is a lack of comparison to relevant baselines. Showing improvements relative to the pretrained model is good but ideally we would see how the method performs compared to other RL methods
- There are well-established issues with FID, particularly using Inception embeddings. In addition, optimizing for the evaluation metric is not ideal. Ideally, DINOV2 embeddings would be used for evaluation (and potentially training as well).
- The writing was hard to follow at times (particularly in the introduction). Improvements could be made to the flow, particularly pertaining to how both methods are presented/motivated.
- The related work would benefit from additional discussion of how RL is used for image generation instead of what felt more like a wall of citations.
- There was a lack of motivation for the issues of sample-wise rewards. While I buy that they are likely more vulnerable to reward hacking, it does not always occur (i.e. not all sample-wise RL methods yields the issues in Figure 1. all the time). Additional theoretical justification could also be given for why the authors expect that this will not be an issue for distribution-wise rewards.

**Questions:**

- I am still confused about the SDE/ODE discrepancy. Why not train using ODE rollouts and evaluate using ODE rollouts? Why is this issue so important that the second method is needed?
- I am less convinced by the model merging results. Improvements seem marginal and the method feels relatively complicated/less-well motivated. Can you provide some context for the scale of improvement here?
	- How does this compare to simpler methods (e.g. grid search) instead of having to train an MLP.
- Did you test any other distribution-wise metric? Could the method be used with diversity metrics for example?
- Have you tested the method on smaller datasets/models? This could be particularly useful for understanding more about the method and performing more detailed ablations.

Overall, I believe the core idea is good (efficient RL training of generative models using distribution metrics) however the paper lacks some crucial aspects. I would be willing to increase my score for some combination of
- Improved writing/structure/justification (particularly for method 2)
- Comparison to relevant RL training baselines
- Investigation of impact of different FID encoders
- Additional discussion of distribution-wise metrics generally

---

### Official Review · Reviewer_CtGX · 2025-10-28

**Soundness:** 2
**Presentation:** 3
**Contribution:** 3
**Rating:** 4
**Confidence:** 4

**Summary:**

This paper proposes distribution-wise reinforcement fine-tuning for visual generative models. The core idea is to optimize FID directly as the reward via a subset-replace strategy that evaluates a partially refreshed reference set, providing denser and cheaper feedback than recomputing FID on 50k samples each step. The approach is validated on ImageNet by fine-tuning SiT ($256^2$) and, in a separate setting, by using the same reward to learn post-hoc checkpoint-merging weights for EDM2 ($512^2$). Empirically, FID-50K drops from $8.30$ to $5.77$ on SiT after 450 RL steps, and improves from $3.74 \rightarrow 3.52$ (XS) and $2.57 \rightarrow 2.52$ (S) on EDM2. A comprehensive ablation suite—varying the reference set, the number of replacements, and the sample-selection policy—demonstrates the approach’s effectiveness. They further mitigate the SDE–ODE discrepancy by proposing RL-driven post-hoc model merging.

**Strengths:**

- The paper clearly formulates the shortcomings of sample-wise rewards—artifact induction and diversity collapse—and backs this up with persuasive qualitative evidence in Fig. 1. Identifying and articulating this failure is valuable and motivates the shift to distribution-wise objectives.

- The subset-replace mechanism provides frequent, distribution-level feedback by updating a small portion of a reference set and recomputing the reward, with periodic refreshes. It’s simple to plug into existing models and, in the reported experiments, yields consistent FID improvements without architectural changes.

**Weaknesses:**

- The paper provides no analysis of the statistical properties of the subset-based FID reward—e.g., its bias/variance relative to full FID-50K—or of credit assignment when a single scalar reward is applied to all images in a replaced subset. As a result, it remains unclear when and why the policy gradient should align with improving the population FID rather than a noisy proxy.

- The method optimizes FID and also evaluates only with FID (FID-50K; FID-5K in ablations). Without complementary metrics (e.g., KID, precision/recall, CLIPScore/FID-CLIP, or a user study), it is hard to rule out metric overfitting. Adding at least one complementary metric would greatly strengthen the claims.

- Figure 1 provides a qualitative comparison between a sample-wise RL baseline and the proposed approach, but there is no quantitative head-to-head evaluation under matched data and compute. This makes it difficult to attribute the reported improvements specifically to the distribution-wise formulation. If the authors’ claim holds, the sample-wise RL baseline should exhibit higher FID due to mode collapse, and demonstrating this empirically would make the argument more convincing.

Despite these weaknesses, I find the core idea of this paper to be novel and meaningful. If the identified issues are addressed convincingly, I would be inclined to raise my score.

**Questions:**

- I was curious about the specific setup used in Fig. 1. What sample-wise reward function and settings were used to produce the “reward hacking” example?

- The stagnation observed under ODE when training with SDE rollouts is quite intriguing to me. Do you have any hypotheses or diagnostic insights that might help explain this behavior? I’m simply curious about how the authors interpret this phenomenon.

- In Tab. 2, I’m curious whether the reported results for EDM2-XS and EDM2-S were obtained using the post-hoc EMA merging strategy proposed in the original EDM2 paper [1], or if standard EMA was applied instead. If it’s the latter, would it be possible to also report the results with post-hoc EMA applied?

[1] Karras, T., Aittala, M., Lehtinen, J., Hellsten, J., Aila, T., & Laine, S. (2024). Analyzing and improving the training dynamics of diffusion models. In Proceedings of the IEEE/CVF Conference on Computer Vision and Pattern Recognition (pp. 24174–24184).

---

### Official Review · Reviewer_ZNAe · 2025-10-31

**Soundness:** 3
**Presentation:** 3
**Contribution:** 3
**Rating:** 4
**Confidence:** 4

**Summary:**

This paper proposes a novel RL framework for visual generative models using distribution-wise rewards. Unlike traditional sample-wise reward approaches, which often suffer from reward hacking and reduced diversity, the proposed method utilizes metrics like FID to assess the generated samples' alignment with the real-world data distribution. The authors introduce a subset-replace strategy to efficiently compute distribution-wise rewards while addressing computational challenges. Additionally, they suggest optimizing post-hoc model merging coefficients to address the train-inference inconsistency caused by SDE in RL training. The proposed method significantly improves generative performance in terms of FID scores and visual quality across various models, inc

**Strengths:**

The use of distribution-wise rewards and the subset-replace strategy are novel, providing a much-needed solution to the common issues in RL-based visual generation. The results across multiple models (SiT, EDM2) and datasets (ImageNet) provide strong empirical evidence for the proposed method's utility.

**Weaknesses:**

While the subset-replace strategy mitigates computational costs, there may still be challenges in applying this approach to very large-scale datasets or more complex models, and this should be discussed in more detail. The approach seems effective for the tested models, but the paper could benefit from further discussion on how well it scales to different types of generative models or to more complex datasets. The post-hoc model merging strategy is promising but might add extra complexity to the overall training pipeline. The paper could provide a more detailed discussion on the potential engineering challenges involved.

**Questions:**

1.How well does the subset-replace strategy scale when applied to large datasets, such as high-resolution images or video data?
2.Could the post-hoc model merging be adapted to other domains or models beyond visual generation?
3.Can the subset-replace strategy be generalized to other generative tasks beyond image generation (e.g., video generation models)?

---

### Note · Authors · 2025-11-14

I have read and agree with the venue's withdrawal policy on behalf of myself and my co-authors.